# Hand Measurement System Based on Haptic and Vision Devices towards Post-Stroke Patients [note 1]

**DOI:** 10.3390/s22052060

**Published:** 2022-03-07

**Authors:** Katarzyna Koter, Martyna Samowicz, Justyna Redlicka, Igor Zubrycki

**Affiliations:** 1Institute of Machine Tools and Production Engineering, Lodz University of Technology, 90-537 Lodz, Poland; 2Institute of Automatic Control, Lodz University of Technology, 90-537 Lodz, Poland; m.samowicz@gmail.com (M.S.); igor.zubrycki@p.lodz.pl (I.Z.); 3Neurological Rehabilitation Municipal Medical Centre, 90-113 Lodz, Poland; justyna.redlicka@umed.lodz.pl

**Keywords:** vision system, haptic technology, leap motion, measurement system, hand plegia, stroke rehabilitation

## Abstract

Diagnostics of a hand requires measurements of kinematics and joint limits. The standard tools for this purpose are manual devices such as goniometers which allow measuring only one joint simultaneously, making the diagnostics time-consuming. The paper presents a system for automatic measurement and computer presentation of essential parameters of a hand. Constructed software uses an integrated vision system, a haptic device for measurement, and has a web-based user interface. The system provides a simplified way to obtain hand parameters, such as hand size, wrist, and finger range of motions, using the homogeneous-matrix-based notation. The haptic device allows for active measurement of the wrist’s range of motion and additional force measurement. A study was conducted to determine the accuracy and repeatability of measurements compared to the gold standard. The system functionality was confirmed on five healthy participants, with results showing comparable results to manual measurements regarding fingers’ lengths. The study showed that the finger’s basic kinematic structure could be measured by a vision system with a mean difference to caliper measurement of 4.5 mm and repeatability with the Standard Deviations up to 0.7 mm. Joint angle limits measurement achieved poorer results with a mean difference to goniometer of 23.6º. Force measurements taken by the haptic device showed the repeatability with a Standard Deviation of 0.7 N. The presented system allows for a unified measurement and a collection of important parameters of a human hand with therapist interface visualization and control with potential use for post-stroke patients’ precise rehabilitation.

## 1. Introduction

The hand measurement is essential in a diagnostic process, allowing forming the basis for precise hand medicine, guiding and personalizing therapy and possibly providing personalized robot parts for rehabilitation. In this case, accurate hand measurements are required to calibrate the machine or even create an effective end effector for particular hand anatomy and consider the patient’s needs and limitations. These possibilities are instrumental in the case of stroke patients who may suffer from paresis of the upper limb. Diseases of the Central Nervous System (CNS) are the most common cause of movement disorders. Stroke is one of the main reasons for hospitalization in Neurological Rehabilitation Units. According to the National Association of Stroke data, 10% of stroke patients recover almost complete motor and mental health, 25% have a minimal impairment, and 40% have a moderate or severe functional and cognitive impairment, so they require rehabilitation [1]. The brain has a unique ability to constantly develop, react to changes and adapt to them—neuroplasticity. Thanks to this ability, human possibilities are enormous. In the event of damage and/or inactivation of specific structures, spontaneous repair changes occur, aiming to reintegrate brain functions. The death of a nerve cell results in the loss of its function, while with appropriate therapy, regeneration and reconstruction of interneuronal connections may occur. The effects of neuroplasticity depend on clinical factors, but also the age, intellect and education of the patient [2]. Rehabilitation after stroke is based on a multidisciplinary, individual approach to problems arising directly from the consequences of stroke and comorbidities in order to enable the best functioning at home and in social life [1].

The functions of the upper limbs are very important for humans. Their lack or limitation (permanent or temporary) is perceived as one of the most severe effects of many pathological changes. The upper limb plays a vital role in everyday activities, such as eating, brushing teeth, or dressing, which affects the patient’s well-being and motivation. The degree of damage affects the functioning, physical activity, and quality of life of patients [3]. Studies suggest that patients with moderate to severe brain injury who received more intensive rehabilitation earlier showed improvement, and earlier rehabilitation was better than delayed treatment [4].

Improvement of the upper limbs, due to the complexity of the functions and various degrees and types of limitations, may be long-lasting and various. At the same time, full recovery cannot be guaranteed. An effective improvement process requires planning treatment stages and selecting treatments. Currently, there are many techniques and methods of rehabilitation. Verifying their effectiveness requires a prior diagnosis of the patient to determine his capabilities and needs. The tests vary depending on the patient’s condition. For individually selected therapy to the patient’s abilities and needs, a basic analysis of the functions of the upper limbs is necessary, which may include [5]:Goniometric measurements;Linear measurements;Strength.

The literature provides the most frequently used research tools, depending on the patient’s needs, functional and psychological state. The therapist’s current examination of the patient consists of a physical examination—interview and physical examination—linear measurements (length, circumference), examination of the range of motion in the joints, or assessment of muscle strength. The tool for conducting the physical examination is usually a centimeter tape or a goniometer [6]. Depending on whether we have a patient with increased muscle tone or with decreased muscle strength, we adjust a specific study [3,7].

The method is cheap, portable, and easy to use. However, different types of joints require different shapes and sizes of goniometers. Only one joint can be measured simultaneously, making the diagnostics time-consuming. In addition, the precision of the measurement depends on the accuracy of the device and the investigator. Manual reading makes the presence of a qualified therapist during the examination necessary.

The necessity of precise hand modeling and diagnostics induced the use of computer vision in the measurement process [8]. Attempts to gesture recognition include methods from 2D locations of fingertips and palms [9], hand blobs analysis with the use of Kalman filter [10] to Intelligent Human–Computer Interaction (IHCI) and Virtual Reality (VR). Human–Computer-Interaction systems develop a Natural User Interface (NUI), which provides better interaction between computer and user, adapting sensors, gestures, and human senses such as touch or vision. Hand modeling and gesture recognition, which found application in many fields, from video games [11] to sign languages [12], are nowadays playing an important role in modern diagnostic techniques and therapy.

The important aspect is the high accuracy of position tracking systems. Novel tracking devices support sub-millimeter accuracy of measurement, such as Polhemus FASTRACK with the positioning precision of 0.76 mm [13]. Methods based on Optotrack or VICON require markers located on the measured hand, which makes the process difficult and time-consuming [14,15]. In addition, accurately positioning markers on a disabled patient can be difficult and affects measurement correctness. A similar problem occurs in the case of diagnostics with a goniometer.

Post-stroke hand state may significantly impact the ability to obtain the hand position necessary for comprehensive measurements of its anatomical dimensions; for example, patients may feel pain when the hand is touched. For that, 3D methods based on Leap Motion and Kinect promise fast, contactless hand tracking and diagnostics. Systems designed for gesture recognition [16,17] do not require separate markers on specific anatomical segments of the hand and allow measurements without touching the device. A non-contact examination is safer for the patient and does not cause pain or discomfort. In addition, it allows conducting measurements for a wide range of patients in compliance with safety and hygiene principles, as the method does not require the disinfection of the device after use.

The measurement capabilities of the two above-mentioned 3D measurement systems are different. According to Raheja [18], Microsoft Kinect allows accurate identification of the center of the palm and fingertips with the hand segmentation process. A similar approach is shown by Lu [19] where positions of fingertips are determined based on the body information, body index information, and depth information from Kinect. More detailed data of hand position, including wrist orientation and positions of the palm and fingers, are possible with Leap Motion [20].

The attempts to hand diagnostics show that it is possible to detect hand postures, including angles between fingers [21], finger abduction angles and metacarpal joint flexion angle [22]. These authors conclude that hand (angle) measurements with Leap Motion are a reliable alternative to goniometer and have a potential for use in clinical and research settings [23]. However, the authors do not provide details on how their measures connect to Leap Motion API values. In addition, the authors do not verify the quality of Leap Motion-provided bone lengths.

Diagnostics of function and ability in the hand, among others, requires muscle strength measurements. For that purpose, dynamometers can be used [24]. Then, muscle strength is measured based on the force generated directly by the hand muscles. Another method is the use of dedicated electronic devices. Hammer et al. described the use of the Grippit device [25] in order to test the maximum and average force that a tested person can exert on the device for 10 s, trying to use the highest gripping force during the entire measurement.

Another equipment used in medical diagnostics and rehabilitation can be a haptic device. One of the features of haptic devices is their direct contact with the user through exerting forces by the device effector. Both patients and doctors can use haptic controllers depending on their purpose and software. Haptic technology can be used to increase the accuracy of surgeon’s movements and minimize postoperative edema [26], teleoperations [27], and for surgical procedures performed in the oral cavity, [28]. It can also be used as an element of virtual reality environment in training systems for future surgeons [29,30]. Haptic technology is also used in rehabilitation. One of the examples is the Phantom Omni [31] haptic device, which was used as a tool to support the re-learning of writing after a stroke. The Phantom Omni haptic device allows to read the effector position with an accuracy of 0.05 mm and exert a force of 3.3 N through the effector. The Phantom haptic device was also used to diagnose and rehabilitate a patient with pronounced paresis of the left hand. For this purpose, an exercise was carried out on the proprietary software cooperating with the Phantom haptic controller and on a particular screen imitating a 3D image.

An essential aspect is the user interface dedicated to measurements. A correctly conducted examination is the basis for determining rehabilitation goals and selecting appropriate therapeutic methods. The repeated examination with the same methods allows controlling the effects of the applied method. To achieve certain goals, proper documentation is needed, to which in practice insufficient attention is paid [5,32]. The compilation and availability of a medical history enable clinicians to compile a comprehensive overview of patient conditions, procedures, medications, family history, and social situations. Therefore, medical data visualization meets with increasing approval among doctors and therapists. Thanks to it, it is much easier to observe the effects of therapy, extract the most important information and draw conclusions from patient data. The development of dashboards and interfaces eliminates time-consuming, manual data analysis and facilitates data reading. Dashboards were used in the evaluation of treatment results as sepsis [33], monitoring rehabilitation progress and exercise management [34] or in systems supporting therapists [35]. Data such as the previous and current hand’s range of motion help check the patient’s condition. Therefore, it is necessary to equip the measurement system with a dedicated user interface to carry out measurements, compare results, and identify and store patient data.

According to Collins et al. [36], a new approach to health care will involve precision medicine. Today, artificial intelligence and sensor-based systems are gaining popularity among the medical community and are increasingly used for monitoring the condition of the patient and adjusting necessary intervention [37,38]. This concept allows for individual treatment and prevention strategies. It can be beneficial in the case of clinical diagnostics, which are often based on the observation of subjects. By providing rehabilitation specialists with tools that allow for fast, accurate measurements, it is possible to adjust the treatment to personal needs. In addition, providing access to databases to monitor patients’ states allows for treatment progress control.

The paper aims to present a system that allows to measure essential parameters of a human hand and to present the results to a therapist. The conducted study aimed to determine the accuracy and repeatability of measurements compared to the gold standard. The designed system consists of Leap Motion, a haptic device, and a user interface. Confirmation of the applicability of the proposed measuring system will allow for further work aimed at creating a complete measurement system for the disabled hand. The proposed system is an improved version of the system based on Leap Motion, which we presented on SPSympo 2011 [39].

### Measurement System

Based on the above review, we focused on creating a system of measurements that would allow accurate measurement of medically essential parameters and the interface for the medical personnel to control the measurement procedure and collect the results. According to Lopatka, the main tests on the upper limb are anthropometric (measurements of length, joint mobility, and muscle strength). The arm, forearm, and all bones of the hand, including the phalanges, are measured [40].

On this basis, we propose the following measurement parameters:Finger and hand sizes;Hand width;Wrist extension-flexion range;Spread of fingers;Finger flexion;Wrist extension-flexion force.

We based our system on a Leap Motion sensor and a haptic device. The designed system allows for collecting geometric parameters of the hand using a non-contact vision system and the measurement of the wrist strength using a haptic sensor. Additionally, the use of commercially available devices means that the measurement procedure does not require the design of a separate measuring device.

The operation of the designed system was verified on healthy users to confirm the correct operation and repeatability of the collected data. This paper provides a concise and precise notation of the measures calculated based on the skeleton data provided by the Leap Motion device and its API, followed by hand measurement via the haptic device. In this research, we compare the Leap Motion results provided by our system to a goniometer (similar to [23]) and extend it also to verify the precision of bone length values by comparing them to caliper measurements. We expanded research with the haptic device’s measurements of wrist range and wrist force while comparing the results with dynamometer measurements.

Effective use of the proposed measurement tools requires a dedicated user interface, which will allow the measurement devices to be coupled into one system. The therapist can select the tested hand parameter and start the measurement from the interface level. It is also important to track the progress of rehabilitation as a result of analyzing the measurements taken during therapy.

The paper describes the components of the system, and the results of the system operation verification carried out on healthy participants. The parts of the system are shown in Figure 1.

## 2. Leap Motion Data Acquisition and Processing

Measuring the geometric parameters of the hand with Leap Motion required the determination of vector coordinates corresponding to the location of individual bones. Then, it was necessary to define the equations describing the relationships between successive vectors to determine the hand lengths of the angles. The data acquisition procedure and method of calculating data from measurements are presented below.

The data acquisition and processing pipeline are shown in Figure 2. The current version of Leap Motion SDK is available only for Windows. Therefore, we used a Python wrapper for Leap Motion 4.0.1 SDK provided by ROSE Motion [41]. We acquire positions of bones and orientation quaternions of the hand and wrist.

We send each LeapMotion frame as a JSON string to ROS (Robot Operating System) using ROS Bridge Websocket connection and decode it using a constructed ROS Node. We then construct a connected graph of coordinate frames (hand skeleton), where each transform frame is described relative to its neighbor that we broadcast to the ROS system using the TF package [42].

A separate node (Python language program) can subscribe to the transform messages and query for particular transformation frames. As the transformation frames form a connected graph, any transform inside a hand can be retrieved easily. The transform frame is described as a homogeneous matrix, i.e., the rotation is described as a rotation matrix, while a position is described as a homogeneous vector, forming a 4×4 matrix. In particular, the position vector describes the center of a transform frame that for most frames, means the beginning of a specific bone (i.e., metacarpal, proximal, distal) while the z-axis lies in parallel to the bone. Because of that, the third column of the rotation matrix (describing the z-axis of the child transform frame described in the parent frame) is of most interest. Figure 3 shows the hand skeleton graph constructed and distributed in the ROS system.

We construct a network of nodes that use particular transforms or sets of transforms, and additionally, we attach a user interface node that can collect data. This approach enables us to run concurrent data processing schemes as separate processes with the ability to switch them on and off when needed.

### 2.1. Measurement Procedure

We can calculate multiple hand measurements using the prepared graph of transform frames and their homogeneous-matrix transformations. The equations describing the dependencies enabling the determination of parameters have been determined for all defined variables of fingers and hand size and motion range.

#### 2.1.1. Finger and Hand Size

For particular hand bone, we calculate the L2 norm of the translation vector, being the first three elements of the last column of the homogeneous matrix constructed between neighboring nodes of the skeleton graph (Equation (Equation 1)).
(1)$$L_{bone}=\sqrt{{(}^{prev.bone}H_{bone}{\left[1:3,4\right]}^{T}{}^{prev.bone}H_{bone}\left[1:4,4\right])}$$
(2)$$HL=\max_{finI,R,M,P}L_{fm}+L_{fp}+L_{fd}$$
where HL is Hand Length, *I*—index, *R*—ring, *M*—middle, *P*—pinky fingers, *m*—metacarpal, *p*—proximal, *d*—distal bone. We find the maximum length out of the finger (and metacarpal bone) lengths. This approach is possible because the hand model used by leap motion does not have carpal bones, i.e., the length is included in a metacarpal bone (Equation (Equation 2)).

For the hand width, we take the absolute value of the first (x-th) element of the translation vector of the transformation frame between the beginning of a proximal frame of index and pinky fingers (Equation (Equation 3)).
(3)$$HW=∥({}^{Pp}H_{Ip}\left[1,4\right])∥$$
where HW is Hand Width, Pp is pinky finger proximal bone frame, Ip is index finger proximal bone frame.

#### 2.1.2. Finger and Hand Motion Range

Wrist angle can be described as a roll angle (rotation around x-axis) of the wrist transformation frame. We calculate roll, pitch, yaw angles from the transformation matrix using pyKLD. During wrist flexion and extension, the algorithm collects extreme values to calculate the full range of wrist motion, using Equation (Equation 4).
(4)WM=minRA−maxRA
where WM is full Wrist Motion range, RA is a wrist-roll angle.

Metacarpal and proximal transformation frames are used to calculate finger spread. Spread angle of each finger is calculated based on the third (**z**) column of the transformation matrix ${}^{fM}H_{fP}$ and its first and third element ($zx={}^{fM}H_{fP}\left[1,3\right]$, $zz={}^{fM}H_{fP}\left[3,3\right]$). Then, the difference of angles between adjacent fingers (except the thumb) is calculated. Total finger spread is the cumulative sum of the differences of angles between the fingers.
(5)$$FSR=\sum_{i=1\ldots 3}\left(atan2\left(zx_{i+1},zz_{i+1}\right)-atan2\left(zx_{i},zz_{i}\right)\right)$$
where FSR is Finger Spread range, *i* is the number of a finger numerated from 1 (pinky) to 4 (index), Equation (Equation 5). Similarly, using the second and third elements of the matrix, flexion and extension angle can be calculated. We call this measure FFR (finger flexion range), Equation (Equation 4).
(6)$$angle_{MP}=atan2\left(M_{MP}\left[2,3\right],M_{MP}\left[3,3\right]\right)$$
where $M_{MP}$ is rotation matrix element of relative frame between metacarpal and proximal joint frames.
(7)$$angle_{TIP}=atan2\left(M_{PTIP}\left[2,3\right],M_{PTIP}\left[3,3\right]\right)$$
where $M_{PTIP}$ is rotation matrix element of relative frame between proximal joint and fingertip frames.
(8)$$FF=max\left(angle_{MP}\right)+max\left(angle_{TIP}\right)$$

Finger flexion can be described as the sum of the maximum of metacarpal-proximal (Equation (Equation 6)) and proximal-fingertip (Equation (Equation 7)) flexion angles. It is calculated according to the formulas (Equation (Equation 8)). To analyze the correctness of the algorithm’s operation, we currently measure the range of finger movement for the index finger only.

## 3. Haptic Device Station

Wrist range and force measurements were measured with the Omega 7 from Force Dimension. It is a haptic device with 7 degrees of freedom. The working space of the Omega 7 is Ø 160 × 110 mm. The rotation allowed by the gripper of the device is 240 × 140 × 180 deg. A button that imitates a grip allows a movement of 25 mm. In the working space, the device can generate forces up to 12 N, while the grip button can generate forces up to 8 N. The accuracy of reading the device’s coordinates is 0.01 mm for moving in the x, y, z axes. For the grip, the accuracy is 0.006 mm, while when it comes to reading the angles, which the gripper tip of the Omega 7 allows, it is 0.09 deg. Communication with the device interface takes place as standard via USB 2.0 and allows for refreshing with a frequency of up to 4 KHz [43].

### 3.1. Data Acquisition

The measuring station equipped with the haptic device was used to measure the maximum wrist deflection angle and the force required to obtain such deflection angles for a patient. Therefore, the method of operation of such a program must be a kind of imitation of the therapist’s activities while working with the examined hand. This means that an important element of the measurement software for the haptic device is the function that measures the wrist deflection and its force. The function considers the initial values of the hand position of the tested person, which are recorded before starting the measurements and immediately after the tested user places his hand in the appropriate position. The function then aims to gradually increase the force tilting the hand, acting perpendicularly to the hand. Then, it measures the angle of tilt on an ongoing basis. The function has a clear signal when it is to stop increasing the tilting force and to obtain adequately this signal, end the tilting and record the values of force and angle obtained during the measurement. For example, the predetermined wrist deflection angle is the signal by which the measurement will be completed. The flowchart in Figure 4 shows the measurement process. In addition to the module responsible for collecting the plegic hand measurements, it is necessary to prepare an algorithm that enforces certain speed limits of the setter effector to prevent possible injuries caused by an acceleration of the system to dangerous speeds. The forearm stabilizing grip and the haptic set must be safe to use, even if one tries to misuse the station. Additional functionality of the software is to keep the haptic setpoint effector at a set height about the handle so that the hand does not fall inertly. Dropping the hand could significantly affect the measurement errors of the force and the deflection angle, which are calculated above the plane.

### 3.2. Measurement Procedure

The measurement performed with a haptic adjuster refers directly to the element to which the hand of the examined person is attached. Coordinates of the space point where the user’s wrist is located and the coordinates of the haptic setter effector to which the hand is attached are required. From these coordinates, two vectors can be determined. Vector A was created from the wrist-effector points, while the hand is in the entry position, and the vectors B and C are vectors made of the coordinates of the wrist and the effector of the setter, while the hand is in two extreme deflections. These vectors are shown in Figure 5 adequately to the wrist bend. The total deflection angle is the sum of the angles between vectors A and B and vectors C and A. Points in the working space of the haptic adjuster are points of three variables, so they contain information about the position in relation to the three axes—x, y, z. The angles should consider all three coordinates of the point or prevent the effector from moving in one of the axes and therefore use points on the x, y plane during calculations. The controller module responsible for the measurements, apart from the deflection angle, is also responsible for generating forces perpendicular to the vector wrist-effector, in such a way that it was possible to test the force that caused the wrist to move to the maximum angle. The algorithm responsible for generating the force gradually increases the generated force until a certain deflection angle is obtained. After reaching a predetermined angle, the force does not increase and acts on the hand until the angle does not change further over time. It is essential for correct force readings that the force vector acting on the effector is always perpendicular to the vectors A, B, and C described above. An exemplary vector of the force tilting the hand is illustrated in Figure 5, and it is marked in yellow. After exceeding the specified deflection angle (in this case, 15 degrees), the program stops increasing the deflecting force and continues to deflect the wrist. It continues to deflect with the force that was set until the angle of 15 degrees is reached for the next “x” seconds. After the time “x” has elapsed, the measurement is saved in the form of three values: the angle of the wrist deflection; the force used to deflect the wrist to a given angle; and the direction of the wrist deflection or the angle at which you want the force to stop growing. One should include in the software an easy way to change these variables.

### 3.3. Forearm Stabilizer

The haptic device used in this project is not directly adapted to work in medical diagnostics, so an appropriate station was prepared to stabilize the patient’s limb under examination. That allows the user to perform diagnostics appropriately using the device. In addition, it allows adjusting the position of the patient’s hand according to the different sizes of the limb. The forearm is placed on a holder equipped with ties stabilizing the hand’s position. The holder is placed on a guide that allows movement along the axis of the device to obtain the appropriate distance between the hand and the device. The holder has been made to be comfortable during more prolonged use and protects the limb from injuries. In addition, the holder socket where the forearm is placed is tilted to the ground by an angle α = 3.5 degrees. This means that the hand attached to the setter effector does not require wrist bending before starting the measurements and allows for free measurement of the movement of the wrist deflection. The measurement station is shown in Figure 6.

## 4. User Interface

The task of the user interface is to enable the measurement of selected parameters with a dedicated device and then read the results. Additional functionality is the ability to view the history of measurements for a selected patient, thanks to which the therapist can control the progress of rehabilitation. The first version of the interface was dedicated to therapists performing the hand measurements with the Leap Motion vision system [39]. We performed a survey for neuro-rehabilitation students (9 respondents) and practitioners (1 respondent) to evaluate the doctor’s interface. Survey questions concerned simplicity of interface, accessibility, and intuitiveness of measuring process and displayed results. Based on the feedback, we improved the interface and added the functionality of haptic measurements.

The schema in Figure 1 shows the data flow used in the study. The whole procedure is taken by the therapist taking the measurement. The therapist uses the presented interface to measure the parameters of the hand using a haptic sensor and a Leap Motion. It includes 4 sections: Welcome Panel (Figure 7a), Leap Motion Measurements (Figure 7b), Haptic Sensor Measurements (Figure 7c), and Patient Panel (Figure 7d).

In the section on measuring with the vision method, we have the option of a basic measurement configuration: hand selection and selection of a specific hand parameter, e.g., hand spread. Additionally, the correct movement is displayed to ensure that the correct measurement is taken. So far, the haptic measurement panel only allows for a simple collection of measurements, but in the future, it is worth visually showing the correct movement during the measurement procedure. Both measuring panels inform the therapist about the necessity to place the hand in the correct starting position. Information about the start of measurements and their completion is displayed. It is worth informing about the progress, duration, and whether the measurement was performed correctly. Ultimately, after each measurement, the results will be visible after a brief analysis.

The patient’s panel allows the therapist to view all the patient’s data from the entire duration of the therapy, both about his data and about therapy indicators. All data are collected in a database. The therapist also can add a new patient in this section.

## 5. Verification Study Materials and Methods

This section contains a study design that we used to assess system validity. The section defines goals and components that have been selected for system implementation and validation. Measurement procedures and statistical analysis methods used for results verification are described. Study participants are described.

### 5.1. Study Design

The designed system required verification of its quality. For this purpose, a series of measurements were carried out on healthy participants, measuring previously selected parameters using Leap Motion and a haptic device. Tests were conducted to determine the accuracy and repeatability of measurements and check what factors should be taken into account to adapt the system to the needs of people with a disabled hand.

The tests of geometric hand parameters were carried out using Leap Motion SDK Orion 4.1.0 described in Section 2 of the paper. Following parameters were measured: hand length and width and lengths of fingers: thumb, index finger, middle finger, ring finger, pinky finger. Angle measurements concerned proximal-tip finger flexion, metacarpal-proximal finger flexion, wrist range, and fingers’ spread.

Wrist range and force were measured using two Omega 7 haptic devices, with handles adapted to the left and right hand. A characteristics of the device and measurement method is described in Section 3 of the paper.

Comparative tests were carried out using a digital caliper, goniometer and dynamometer. Lengths of the hand and fingers were measured by an electronic digital caliper calibrated for the 0–150 mm scale. The goniometer with the range of 360º and scale increment 1º was used for angle measurements. Wrist strength test was performed with dynamometer with the range of 0–10 N and scale increment 0.2 N.

### 5.2. Study Participants

The study was conducted for five participants, at average age 22 ± 1. There were four female and one male participant. All participants were healthy with no hand disabilities. All study participants gave their informed consent to participate in the research. In addition, we obtained ethical approval of the study from the Bioethics Committee at the Medical University in Lodz.

### 5.3. Measurement Procedure

For assessments of the skeleton and hand flexion hand parameters with Leap Motion, each participant was asked to move fingers and wrist actively. Participants were also asked to perform the hand’s maximum movement for the flexion measurements, expand fingers and the wrist, and spread fingers. Hand parameters were calculated based on the equations provided in Section 2.1 of the paper. The range of wrist motion and the force needed for hand flexion and extension (Figure 8) were measured using a haptic device while the forearm was stabilized in the holder. A dynamometer measurement of extension/flexion forces was performed by pulling the dynamometer with a uniform force following the bending angle. The measurement value was read when the wrist no longer changed position and felt the resistance. All tests were performed for both hands. Measurement of each hand parameter was repeated 15 times for each hand parameter of all participants.

### 5.4. Manual Measurements

Manual measurements of the hands were made using an electronic caliper and a goniometer. All manual measurements were made according to the instructions in Zembaty’s manual [32]. Detailed descriptions of the measurement of individual hand parameters are presented in Table A1.

A digital caliper was used to measure the hand’s length, fingers, and hand width. The length of the fingers was measured on the straightened hand, between the center of the metacarpal point of the selected finger and its tip. The hand’s width was measured between the metacarpal pinky points of the finger and the index finger. The hand’s length measured from the metacarpal to the tip of the most extended finger.

Hand flexion research includes the range of the wrist motion, the range of the index finger motion, and the maximum range of finger spread. Manual finger spread measurement involves measuring the angle between the pinky and middle metacarpals at the center point of the hand. Tests with goniometer required performing hand poses of maximum flexion by the participants. The range of wrist motion was tested by bending the hand of the examined person by the maximum angle and then measuring the endpoints for both bend angles and summing the results. Finger flexion requires measuring the angles between the metacarpal and proximal on the index finger and the angle between proximal and tip.

### 5.5. Statistical Analyses

To determine the quality parameters of the measurements, we have set the Standard Deviations and Difference of Means for individual values measured using selected devices.

Standard deviations (SD) were calculated based on measurements of all participants while using specified measuring devices, for different hand parts, including the number of samples conducted in each measurement based on Equation (Equation 9).
(9)$$SD_{p,m,i}={\frac{1}{N}}_{p}\sum_{p=1}^{N_{p}}\sqrt{\frac{\sum_{p_{d,h,i}=1}^{n}{\left(x_{d,h,i}-{\bar{x}}_{d,h,i}\right)}^{2}}{N_{d,h,i}-1}}$$
where ($N_{p}$) is the number of participants (*p*), *x* is measured value, $\bar{x}$ is an average value from measurement, *d* is the type of measuring device, *h* is hand part, *i* is number of sample in each measurement.

Difference of means (DM) was determined based on the difference between the average measurement result carried out for all participants with Leap Motion and other measuring devices, for individual parts of the hand using Equation (Equation 10).
(10)$$\;DM_{p}={\frac{1}{N}}_{p}\sum_{p=1}^{N_{p}}\left|({\bar{x}}_{LM,h}-{\bar{x}}_{d,h})\right|$$
where $N_{p}$ is the number of participants (*p*), ${\bar{x}}_{LM}$ is an average measurement with Leap Motion, ${\bar{x}}_{d}$ is an average measurement with other devices, *h* is hand part.

Pearson correlation coefficient was calculated via a web calculator. The correlation was tested for pairs Leap Motion and caliper and Leap Motion and goniometer. In the tests, we used all measurements data containing particular parameters: length, angle.

## 6. Results

Wrist forces required for flexion and extension were measured via a haptic device and dynamometer. All the hand flexion and skeleton data were collected with Leap Motion, Caliper, and Goniometer with supplementary measurement of wrist range via the haptic device. This section presents the results of the geometric and force measurements.

### 6.1. Length and Angle Measurements

This section presents the results of geometric measurements containing length and angle. The statistical analysis allowed for setting Standard Deviation and difference of means between average values. Standard deviations for length measurements by Leap Motion and Caliper and the difference of means between average length measurements are shown in Table 1.

The standard deviation values obtained with the Leap Motion and caliper measurements are comparable and belong to the same series of values. The smallest SD values for Leap Motion were obtained for the measurement of the thumb length SD = 0.7 mm and the length of the little finger and the width of the hand, both equal to 1 mm. The highest standard deviation value with Leap Motion measurements was obtained for the hand length SD = 1.8 mm. The values of the standard deviation obtained with the caliper were slightly lower and reached the maximum value for the measurement of the length of the thumb and little finger, SD = 0.9 mm, and the lowest for the measurement of the length of the middle and ring fingers, SD = 0.6 mm. For Difference of Means, the minor difference in obtained measurement values was observed in the case of measuring the length of the little finger, equal to 4.5 mm. The most significant difference in measurements was obtained for the hand length measurement and was equal to 19.1 mm. Considering the high repeatability of the measurement achieved with both methods, the differences in the obtained measurements may result from difficulties in determining the same reference points. The Leap Motion measurement is based on the measurement of bone length. In contrast, in the case of a caliper, the measurement is also influenced by other structures of the human hand, which may cause discrepancies in the results obtained.

Standard deviations for angle measurements and the difference of means between average measurements are shown in Table 2. In the case of measuring angles, the values of the standard deviation were compared based on the results obtained using three measuring devices—Leap Motion, haptic device and goniometer. The haptic device measurements allowed us to obtain the lowest value of the standard deviation among the available measurement methods. The haptic device measurements of wrist range were characterized by high repeatability, with SD = 4.0º. The lowest repeatability was achieved for the wrist range measurement with Leap Motion, SD = 12.3º. In addition, measurements of angular values showed that Leap Motion achieves lower repeatability of the measurement than measurements with a goniometer. For finger flexion, SD = 16.5º was obtained for Leap Motion, while for the goniometer SD, it was 3.8º. The most comparable standard deviation values among measured angular values were obtained for the fingers spread measurement. For Leap Motion the SD was equal to 7.6º, and for the SD goniometer equal to 2.1º.

Figure 9 shows the comparison of wrist range measurement results for three methods: Leap Motion, Goniometer and Haptic device. The results for one participant are shown with the left and right-hand examples. Based on the presented data, it can be observed that the values obtained in the measurement of the right-hand range are similar, respectively, 128.8º for the Leap Motion measurement and 131.1º for the measurement with the haptic controller were obtained. The goniometer measurement showed a higher value; for the right hand, it is 158.2º. Similarly, for the left hand, it can be seen that the goniometer measurement gave a higher result than Leap Motion and the haptic sensor. The Leap Motion measurement is characterized by the highest standard deviation value among the compared methods.

Standard deviations for measurements by Leap Motion, Goniometer and Caliper and the difference of means between average measurements are shown in Table 3. Additionally, the obtained values of standard deviation and Difference of Means were compared with the previous results of measurements carried out with the use of the older version of Leap Motion, goniometer, and calipers [39]. In length measurements, the current version of Leap Motion allowed for higher repeatability of measurements than the older one. In the case of the index finger, the result was SD = 3.0 mm, while in the current measurements, it was SD = 1.2 mm. Additionally, the repeatability of the caliper measurement has increased. For example, the standard deviation obtained for the middle finger length measurement decreased from 1.2 mm to 0.6 mm. Higher repeatability of the caliper measurement may result from the experience of the measuring person and has an impact on the obtained measurement values. In the case of measuring angles using Leap Motion, it was possible to obtain similar repeatability of the measurement for wrist range and finger flexion.

Comparison of average results of length measurements for one participant are shown in Figure 10. The graph compares the results obtained with Leap Motion and the calipers for fingers: pinky, ring middle and index. The standard deviation values for each measurement method are plotted on the graph. The presented results show discrepancies in the measurement values obtained for each method. The most similar values were obtained for the ring finger measurement. The average length of a participant’s finger was 73.3 mm for the Leap Motion measurement and 76.7 mm for the caliper measurement. Additionally, it can be observed that the caliper measurement results have higher values than those made by Leap Motion. This may be due to the differences between the two measurement methods and designated reference points.

Additionally, the correlation of the measurement methods was tested for caliper, goniometer and Leap motion by calculating the Pearson correlation coefficient. Comparison of linear measurements taken by Leap Motion and Caliper showed a strong positive correlation with R = 0.9695. A strong positive correlation was also observed for angular measurements using Leap Motion and Goniometer, with R = 0.8553. Graphical comparison interpretation is shown on the example of the results for the index finger and hand length. It is presented in the graph shown in Figure 11. On the example of the index finger, it can be noticed that for values over 70 mm in the case of Leap Motion measurement and values over 80 mm for a caliper, the relationship between the obtained results assumes a function close to a linear one. A similar dependence can be noticed for hand length measurements, for Leap Motion measurements above 145 mm and caliper measurements above 150 mm. There is no linear dependence for the lower measured values in both cases.

### 6.2. Force Measurements

The force needed for the wrist flexion and extension was measured using a haptic adjuster and a dynameter. The comparison of the standard deviation values for the series of measurements and the Difference of Means are shown in Table 4. Based on the presented data, it can be seen that the haptic device measurements were characterized by a higher standard deviation, both for flexion SD = 0.7 N and extension SD = 0.6 N. For the dynamometer, the obtained values were SD = 0.3 N for flexion and, SD = 2 N for extension. The difference of means of both measurements is similar and is 0.6 N for flexion and 0.7 N for the extension. Additionally, Figure 12 shows a comparison of the average values of forces needed for flexion and extension for one participant. The results obtained are comparable; for example, the force required for flexion of the left hand is 2.6 N when measured with the haptic device and 2.4 N when measured with a dynamometer. In most cases, the result obtained with the dynamometer and the standard deviation falls within the values of the standard deviations obtained for the haptic device.

## 7. Discussion

### 7.1. Healthy Participants Verification of Measurement System

The results obtained by Leap Motion were comparable to the goniometer and caliper hand measurement methods. It is possible to measure both the length of the fingers and the hand and the range of hand motion. The measurement results showed the highest accuracy when measuring pinky finger length. For this value, the Difference of Means between Leap Motion and caliper is 4.5 mm. Additionally, the obtained results of Leap Motion, Goniometer and Caliper measurements are characterized by high repeatability with the best results for thumb length for Leap Motion and middle and ring finger length for caliper.

Inaccuracies in Leap Motion’s angle measurements made the repeatability of Leap Motion lower than by goniometer. For the finger flexion, Standard Deviation of Leap Motion was 16.5 mm, while for the goniometer measurement, the obtained value of SD was equal to 3.8 mm. A similar result was observed for the wrist range, where the SD values were 12.3 mm for Leap Motion and 6.0 mm for the goniometer. Therefore, we can conclude that the goniometer provides better wrist and finger flexion results and the Leap Motion system would need to be improved. The most repetitive results of wrist range were obtained with haptic device, where SD = 4.0 mm. The force measurements carried out with the haptic device showed that it is possible to measure the wrist flexion and extension force with a parallel wrist range measurement. For force measurement using the haptic device, SD = 0.7 N for flexion and 0.6 N for extension was obtained, resulting in higher values than in the case of dynamometer measurements and may indicate the need to introduce corrections in the measurement system.

### 7.2. Design of the System

The increasing interest of the medical community in sensor-based systems [36,37] means that the system can find medical applications, especially in the case of hand diagnostics after a stroke, and be beneficial in case of the progress of treatment. The work of the presented measuring system has been confirmed by studies conducted on a group of healthy users. The research was carried out on a small group of participants in order to confirm the possibility of integrating the proposed measuring devices into one system, which in the long term will be tested in a larger test group. The work aimed to analyze the accuracy of the measurement sensors with which Leap Motion and the haptic device are equipped and determine the measuring system’s properties. Measurements were carried out for a large number of characteristic parameters of the human hand to accurately identify the system’s measurement capabilities. When verifying the operation of the device, we paid attention to measurement parameters such as accuracy and repeatability.

The use of Leap Motion allows for easy and repeatable measurement of bone sizes and range of hand movement. Unlike manual methods, where it is necessary to use different measuring devices depending on the measured hand part, Leap Motion enables the required measurements to be carried out with one device. In addition, the proposed method is non-contact, which is beneficial due to the hygiene and safety of the patient. Although there are other methods of contactless measurement devices, such as Kinect, Leap Motion allows recognition of fine hand and finger movements [44] and is preferred by medical specialists, especially for measurement tasks [45,46].

Adding the wrist range measurement with the haptic device to the system allows one to supplement the measurement with important parameters needed in diagnostics and rehabilitation. Particularly compared to vision system, the device provides active measurement in which hand can be moved in a safe way. This is important for measurements of patients with plegic or paretic hands. With the pure vision system, therapists assistance would be needed to actuate the hand but this could also occlude the view from the camera (i.e., by occluding patients hand with therapist’s hand) [47].

As the haptic device measurement requires the effector to be grasped by the examined person, further work will be carried out on adapting the haptic device to diagnose people with a disabled upper limb. It can be concluded that the most significant limitation of the current research is the lack of evaluation of patients with post-stroke patients. This will be done in the next phase of the study. In order to obtain the full functional parameters of the hand, it is also necessary to measure additional parameters, such as grip strength, wrist damping, and stiffness. It is possible to modify the work of the haptic device by defining the required bending angle and the force acting on the hand or changing the motion plane, which allows the measurement of additional hand parameters. Further research is being carried out to extend the scope of the parameters studied.

The user interface allows intuitive measurements and provides access to measurement data, which help monitor rehabilitation progress and the patient’s condition. The doctor’s panel allows for Leap Motion and haptic measurements, instructing the therapist about correct hand position. The patient’s panel allows the therapist to view all the patient’s data from the entire duration of the therapy. In addition, it allows for tracking the results of the therapeutic methods used and reacting when, for example, there is no progress in improvement. Extending the available data with the history of the disease or psychological, speech therapy, or other observations will allow looking at the patient comprehensively. All data will be available in one place for medical personnel. As a result, the medical community can find the proposed solution an attractive and objective assessment of the patient’s condition.

## 8. Conclusions

The presented measuring system allows for a collection of important geometric measurements of the hand and flexion/extension wrist force in a unified way. The system’s validation study was carried out on a small group of healthy participants to confirm the possibility of integrating the proposed measuring devices into one system. The study showed comparable results to manual measurements regarding fingers’ lengths. The study showed that the finger’s basic kinematic structure could be measured by a vision system with a mean difference to caliper measurement of 4.5 mm and repeatability with the Standard Deviations up to 0.7 mm. Joint angle limits measurement achieved poorer results with a mean difference to goniometer of 23.6º. Force measurements taken by the haptic device showed repeatability with a Standard Deviation of 0.7 N. Current results show the potential of the system towards measurements of people with disabled hands. The user interface allows the therapist to efficiently carry out measurements and view the results. The use of commercially available measurement devices: integrated vision system (Leap Motion) and an Omega-7 haptic device allow for easier system replication. As the future main goal, we plan to adapt the setup to the needs of people with disabilities and expand the scope of the measured parameters mainly towards the dynamic parameters of the hand.

## Figures and Tables

**Figure 1 sensors-22-02060-f001:**
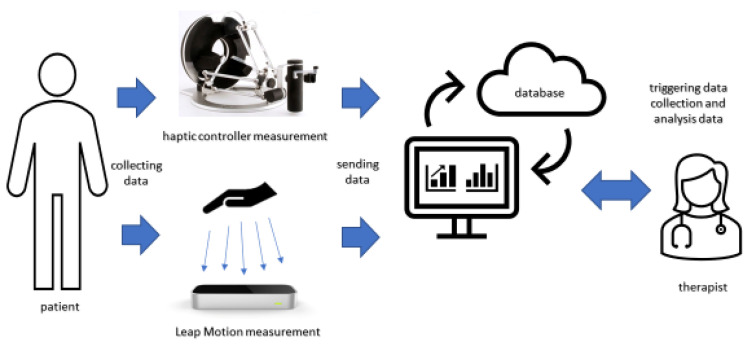
Elements of measurement system based on haptic and vision devices.

**Figure 2 sensors-22-02060-f002:**
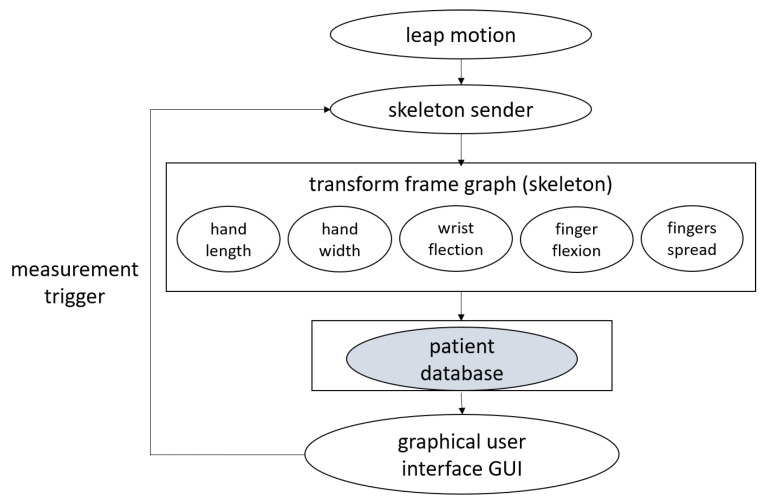
Leap Motion data acquisition and processing scheme.

**Figure 3 sensors-22-02060-f003:**
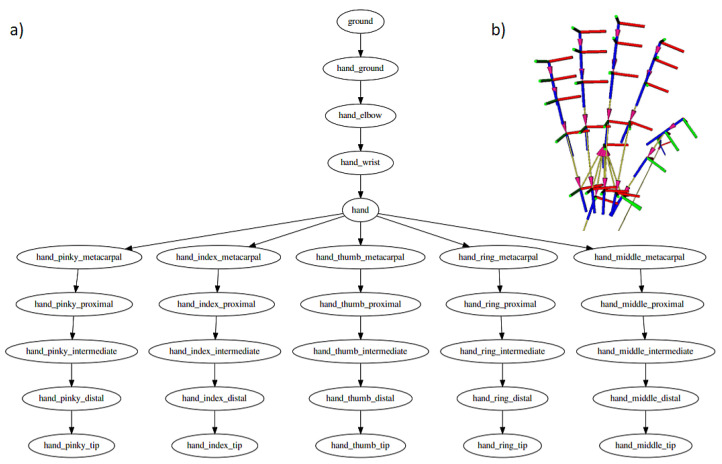
(**a**) Joint connections scheme distributed in the ROS system, (**b**) visualization of bones positions and orientation quaternions of the hand and wrist using ROS system.

**Figure 4 sensors-22-02060-f004:**
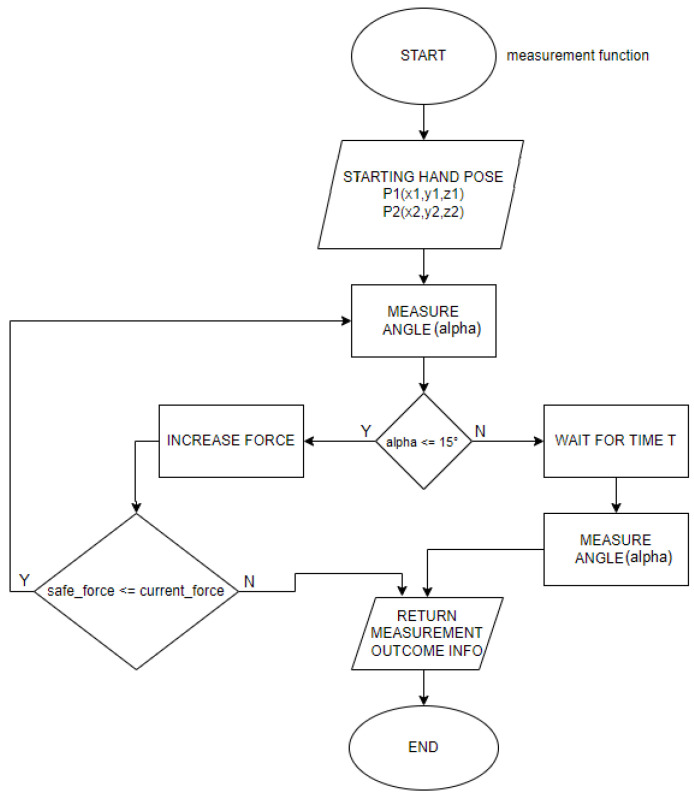
Visualization of user interface showing measurement procedure.

**Figure 5 sensors-22-02060-f005:**
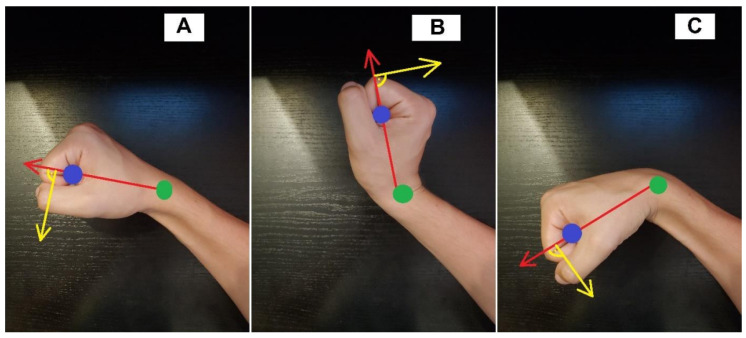
Wrist positions with vectors plotted against which wrist flexion/extension angle was measured. (**A**) vector created while the hand is in the entry position, (**B**) vector created for the wrist extension, (**C**) vector created for the wrist flexion.

**Figure 6 sensors-22-02060-f006:**
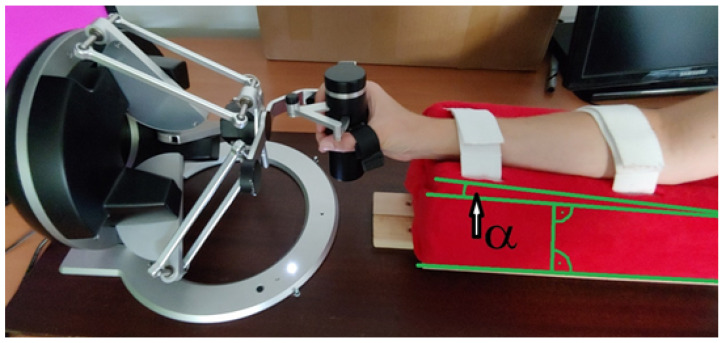
Measurement station with forearm stabilizer.

**Figure 7 sensors-22-02060-f007:**
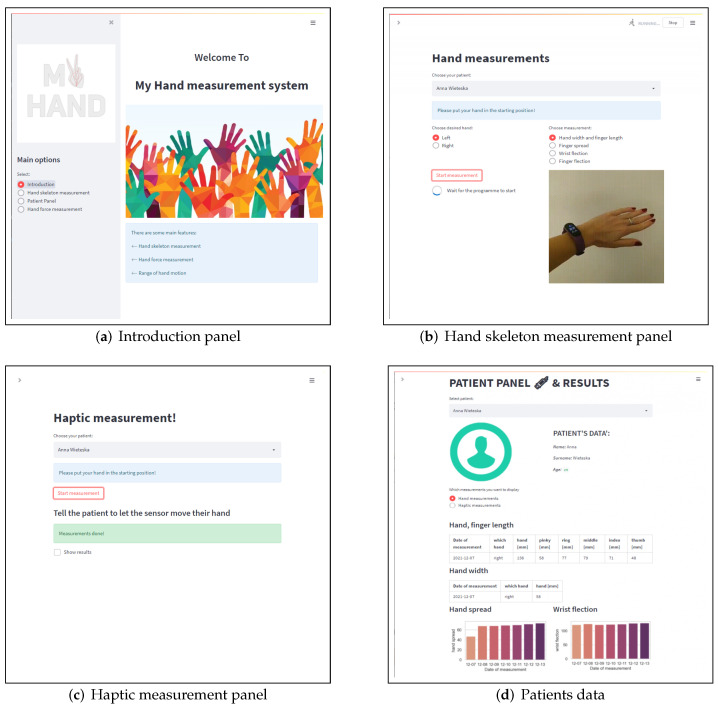
Measurement interface for therapist. (**a**) Introduction panel which choice of the feature section; (**b**) hand skeleton measurement panel for Leap motion measurements, it is possible to choose specific hand parameters; (**c**) haptic measurement panel to collect data of wrist force and range; (**d**) patient’s panel with data from the entire duration of the therapy.

**Figure 8 sensors-22-02060-f008:**
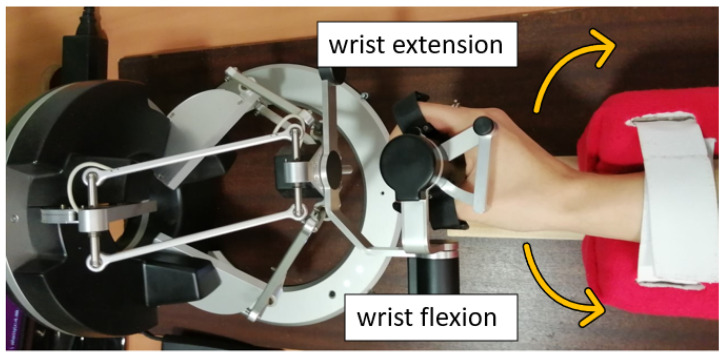
Interpretation of flexion and extension direction on the example of the right hand.

**Figure 9 sensors-22-02060-f009:**
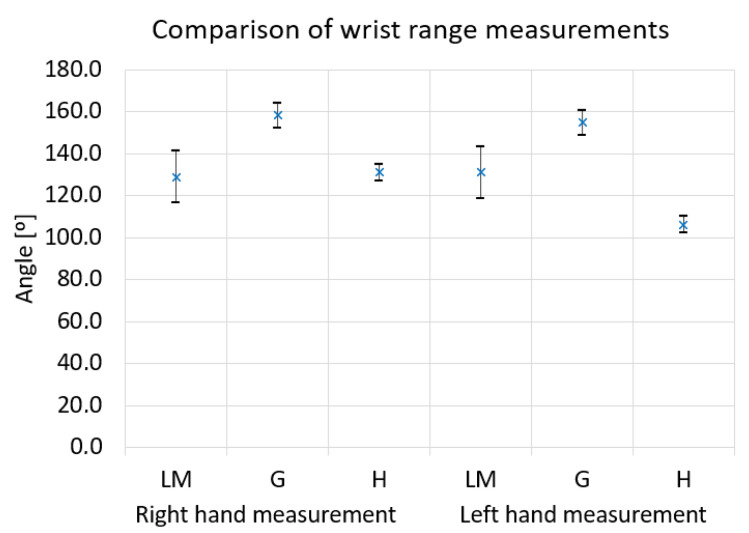
Comparison of angle measurement by Leap Motion (LM) and Goniometer(G) and Haptic device (H) for wrist range.

**Figure 10 sensors-22-02060-f010:**
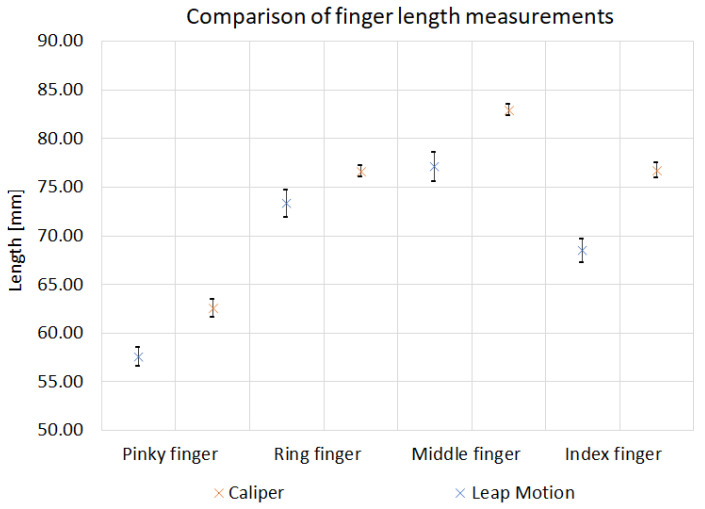
Comparison of finger length measurement by Leap Motion and Caliper.

**Figure 11 sensors-22-02060-f011:**
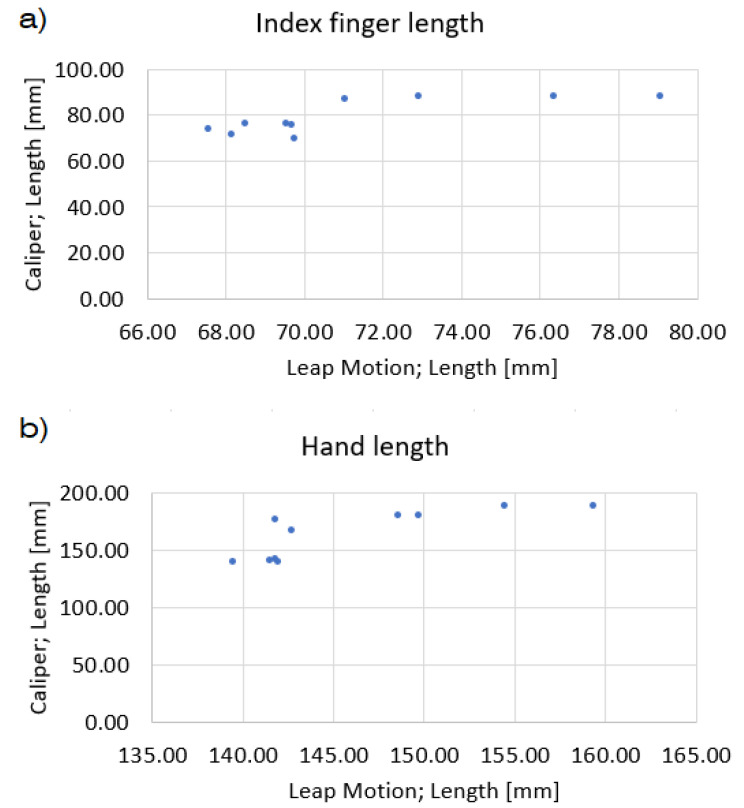
Results correlation of measurements via Leap Motion and Caliper for: (**a**) index finger length, (**b**) hand length.

**Figure 12 sensors-22-02060-f012:**
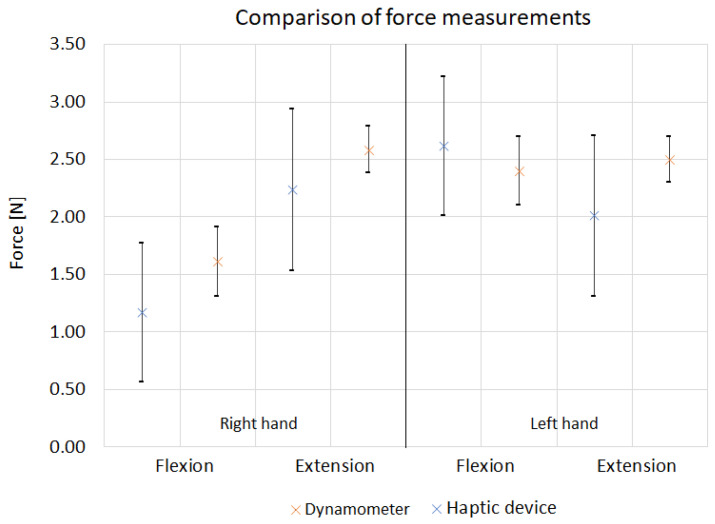
Comparison of force measurements via Haptic device and Dynamometer.

**Table 1 sensors-22-02060-t001:** Standard deviations (SD) and Difference of Means of length measurements.

Measured Value [mm]	Leap Motion SD	Caliper SD	Difference of Means
Thumb length	0.7	0.9	10.3
Index finger length	1.2	0.8	8.6
Middle finger length	1.5	0.6	7.7
Ring finger length	1.4	0.6	6.1
Pinky finger length	1.0	0.9	4.5
Hand length	1.8	0.7	19.1
Hand width	1.0	0.7	9.4

**Table 2 sensors-22-02060-t002:** Standard deviations (SD) of angle measurements.

Angle of: [°]	Leap Motion SD	Goniometer SD	Haptic Device SD
Finger flexion proximal-tip	16.5	3.8	-
Finger flexion metacarpal-proximal	10.7.3	4.3	-
Wrist range	12.3	6.0	4.0
Fingers spread	7.6	2.1	-

**Table 3 sensors-22-02060-t003:** Standard deviations (SD) and Difference of Means of measurements by Leap Motion, Caliper, and Goniometer [39].

	Leap Motion SD	Goniometer SD	Caliper SD	Difference of Means [39]
Thumb length [mm]	2.0	-	1.7	6.7
Index finger length [mm]	3.0	-	1.4	4.9
Middle finger length [mm]	3.3	-	1.2	3.5
Ring finger length [mm]	3.0	-	1.1	3.7
Pinky finger length [mm]	2.5	-	1.4	3.8
Hand length [mm]	5.7	-	4.4	0.5
Hand width [mm]	1.8	-	0.6	1.5
Fingers spread [°]	1.3	2.0	-	1.1
Wrist range [°]	11.3	4.8	-	5.6
Finger flexion [°]	16.6	5.9	-	23.6

**Table 4 sensors-22-02060-t004:** Standard deviations (SD) and Difference of Means of force measurements.

Wrist Force [N]	Haptic Device SD	Dynamometer SD	Difference of Means
Flexion	0.7	0.3	0.6
Extension	0.6	0.2	0.7

## Data Availability

The data presented in this study are available on request from the corresponding author. The data are not publicly available due to privacy.

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
