# Peer review of "Hand Measurement System Based on Haptic and Vision Devices towards Post-Stroke Patientsâ€"

_sensors, 2022, doi:10.3390/s22052060_

Round 1
Reviewer 1 Report
First, I like the idea of evaluating post stroke rehabilitation via hand measurement through the combination of haptic device and vision sensor. Seems very promising and of interest both to academic and industrial research. However, based on the demonstrated results, I don't see the advantage of the proposed measurement system compared to existing systems such as Microsoft Kinect.
Second, the figures shown in this paper are very confusing. Manual measurement of a researcher's hand is not illustrative. The presented data is very scattered and is not telling a compelling story. Makes me lose the interest to finish reading.
I suggest the authors reorganize their story in a concise but explanative manner, demonstrate only useful information and provide strong, meaningful conclusion with evidence of contributions unique to their work.
Reviewer 2 Report
General comment:
The aim of the study is to presents a measurement system for patients with stroke based on an integrated system – Leap Motion and a haptic device. The topic seems relevant but there are some severe limitations to the study that largely reduce the impact. My main criticisms regards the ambiguity in the methodology and lack of solid grounds for some statements, the terminology used, and conclusion drawn. Moreover, the paper has some grammar and language issues that reduces the readability. These limitations lessen the credibility and result in potential study bias. I see no way that the authors could appropriately address these limitations. They firstly need to scale up the study and use a different approach. Therefore, I believe that the manuscript should not achieve sufficient priority to merit acceptance, but should be rejected.
- Firstly, the title is misleading, giving the impression that the authors have included patients with stroke in the study. Instead, assessments were performed on five healthy participants, at the age of 20 to 30.
- The introduction is largely focused on stroke and related disorders. For the reader, it become clear that the authors of the present manuscript do not have any experience in the stroke field or from stroke rehabilitation. As an example, the authors state “The mobility impairment following a stroke is permanent and causes disability”, which is only partly true. Nowadays, it is well-known that patients have capacity to regain function as a result of the brain’s plastic properties.
- The manuscript consistently lacks structure. Nowhere in the abstract or manuscript it becomes really clear to the reader what study design and participants are used. The authors present a measurement system that could be used in clinical contexts to measure force and range of motion in the upper limb in disabled patients.
- The number of participants are too low to be able to draw any valid conclusions.
- Abstract: Change sentence construction into: Patients with stroke are commonly subjected to a series of tests….
- I believe the notion “after-stroke patient” is superfluous. I suggest to simply write “patients with stroke”.
- I suggest to remove the first sentence in the introduction: “This paper presents a measurement system based on Leap Motion Vision Sensor and Omega 7 Haptic Device for after-stroke patients”, since it is completely non-substantiated. Instead, introduce the reader to the field, and subsequently, present the aim with the study.
Round 2
Reviewer 1 Report
I believe the authors have largely addressed my concerns in the revised manuscript and the current version is acceptable if other reviewers don't hold different opinions.
Reviewer 2 Report
My main criticisms still regards the ambiguity in the methodology and lack of structure and solid grounds for some statements, and conclusion drawn. Moreover, the paper has some grammar and language issues that reduces the readability. These limitations lessen the credibility and result in potential study bias. I see no way that the authors could appropriately address these limitations. They firstly need to scale up the study and use a different approach. Therefore, I believe that the manuscript should not achieve sufficient priority to merit acceptance, but should be rejected.
1. Maybe you’re allowed to include an unstructured abstract…? However, I think it largely reduces the readability. In line with the manuscript as a whole, I suggest you structure the abstract differently (background, aim, methods, results and conclusion). Please describe the aim more clearly, i.e. The aim with the present study was to….. Thereafter, describe the methods used and results achieved and what conclusions could be drawn from your findings. The aim in the abstract should be consistent with the aim in the introduction.
2. I’m still not satisfied with the way the manuscript is structured. The aim is written in the first paragraph of the introduction. It should be presented in the last paragraph. The background should turn into the purpose rather than the other way around.
3. I am not satisfied with the presentation of the aim “The paper aims to present a system that allows to measure essential parameters of a human hand and to present the results to a therapist”… The manuscript does present the design of the measurement device but the study also aimed to determine the accuracy and repeatability of measurements as compared to golden standard.
4. Instead of using the title “Measurement principle” you should have a paragraph entitled Study participants, in which you in more detail describe the participants including gender, mean age (SD). Suggest to use the title “Procedure” rather than Mesurement principle.
5. I find no information about ethical approval of the study. This should be included.
6. There is no section Statistical analyses. Instead such information is interspersed in various sections, causing further disorder/disorganization. There is a great need of better structure and specific section relating to study design, participants, procedure, measurements, statistical analyses etc.
Round 3
Reviewer 2 Report
I'm satisfied with the changes and additions made by authors.